# Disease Management Program in patients with type 2 diabetes mellitus, long-term results of the early and established program cohort: A population-based retrospective cohort study

Regina Riedl[1], Martin Robausch[2], Andrea Berghold[1] *

1 Institute for Medical Informatics, Statistics and Documentation, Medical University of Graz, Graz, Austria,
2 Controlling Department (ÄIRCON), Lower Austria Health Insurance Fund, St. Pölten, Austria

* andrea.berghold@medunigraz.at

**Data Availability Statement:** Data cannot be shared publicly due to restrictions from the Austrian Health Insurance Fund. Others can access

## Abstract

### Background

The aim of this study was to evaluate the long-term follow up of the early and the effectiveness of the established program of the Austrian Disease Management Program (DMP) 'Therapie aktiv—Diabetes im Griff' for patients with type 2 diabetes mellitus concerning mortality, major macrovascular complications, costs and process quality of care parameters.

### Methods

We conducted a propensity score matched cohort study based on routine health insurance data for type 2 diabetic patients. The observational period from the matched early program cohort starts from January 1, 2009 to December 31, 2017 and includes 7181 DMP participants and 21543 non-participants. In the established matched program cohort, 3087 DMP participants and 9261 non-participants were observed within January 1, 2014 to December 31, 2017.

### Results

In the early program cohort, 22.1% of the patients in the DMP-group and 29.7% in the control-group died after 8 years follow-up (HR = 0.70; 95% CI: 0.66–0.73). A difference of € 1070 (95% bootstrap-T interval: € 723 - € 1412) in mean total costs per year was observed. In the established program cohort, 10.4% DMP participants died 4 years after enrollment, whereas in the control-group 11.9% of the patients died (HR = 0.88, 95% CI: 0.78–0.99). Healthcare utilization is higher in the DMP-group (75%-96%) compared to the control-group (63%-90%).

### Conclusions

The 8-year long-term follow up of the DMP program showed a relevant improvement of survival and healthcare costs of patients with type 2 diabetes. The established program cohort

these data by applying to the Austrian Health Insurance Fund (vm3-12@oegk.at).

**Funding:** This study was funded by the Styrian Health Insurance Fund (since 2020 Austrian Health Insurance Fund (ÖGK)) in charge of the DMP "Therapie aktiv – Diabetes im Griff". The funders had no role in study design, data collection and analysis, decision to publish, or preparation of the manuscript.

**Competing interests:** Martin Robausch was project leader of the LEICON database, working at the Lower Austria Health Insurance Fund (since 2020 Austrian Health Insurance Fund (ÖGK)), who was responsible for the implementation of the DMP in Lower Austria. This does not alter our adherence to PLOS ONE policies on sharing data and materials. All other authors declare that they have no conflict of interest."

had improved survival and quality of care. Our findings indicate that the DMP "Therapie aktiv" provides a long-term advantage for type 2 diabetes patients.

# Introduction

Health systems need to provide effective care for an increasing number of chronically ill people posing a pressing global health care challenge on the health systems. Especially diabetes mellitus is a major challenge with worldwide about 537 million people aged 20–79 years (10.5% of the population) suffering from diabetes mellitus [1, 2]. In Austria, there are about 515,000–809,000 patients with diabetes mellitus, representing 7% - 11% of the Austrian population [3]. In the last decades, disease management programs (DMPs) and patient self-management programs have been developed to improve health outcomes, quality of life and to reduce the economic impact of type 2 diabetes mellitus [4–6].

Although, for diabetic patients, such programs resulted in slightly better glycemic control and processes of care [7–11], inconsistent results are observed for mortality, risk of macro- or microvascular complications and quality of life compared to usual care [7, 8, 12–14].

In Austria, the Disease Management Program "Therapie aktiv–Diabetes im Griff" (http://www.therapie-aktiv.at) is a systematic treatment program for patients with diabetes mellitus type 2. The DMP was implemented in 2007 based on optional and free of charge participation of general practitioners and patients. A basic training for physicians based on evidence-based clinical guidelines is necessary before they can work as DMP-physicians. Key features of the program include: necessary medical examinations on a regular basis, patient empowerment (by definition of individual target agreements by patient and the physician), lifestyle advice (e.g. change in diet habits and physical activity), and regular medical documentation by DMP-physicians (medical parameters, treatment, target agreements and quality of life). Currently, about 105,530 diabetic patients and 1974 physicians participate in the program (01.10.2022). An evaluation of the early program phase including patients enrolled in the DMP during 2008/2009 showed a lower mortality rate and lower costs compared to patients with usual care after 4 years follow-up [15]. In other early evaluations of the program, an improvement in the quality of outpatient care and lower hospitalization rates for DMP participants compared to non-participants were observed [11]. However, no difference in health related quality of life and a minor effect on metabolic control were observed [13, 16, 17].

To provide further insights about the effectiveness of the Austrian DMP, an observational study was conducted (i) investigating the long-term effects of the program for participants enrolled in the early program phase (DMP enrolment 2008/2009) and (ii) evaluating a cohort of participants enrolled in the established program phase (DMP enrolment 2013). We therefore updated the results for the existing cohort analyzed in Riedl et al [15] increasing the follow-up from 4 years to 8 years. Furthermore, we evaluated the DMP for participants enrolled during 2013 considering patient-relevant outcomes (overall mortality, cardiovascular disease), economic impact and quality of care parameters (e.g. physician contacts, laboratory parameter testing) over a 4 years follow-up.

# Materials and methods

## Study design and data

A retrospective cohort study using a propensity score (PS) matched control-group design was performed to analyze the early and the established program phase. The evaluation is based on

routine health insurance data in accordance with the Austrian general social insurance law, which allows for the use of such data for these purposes (LEICON database). In the database patients with type 2 diabetes mellitus are identified by their form of antidiabetic drug therapy as follows:

- oral antidiabetic drug therapy (OAD) (Anatomical Therapeutic Chemical (ATC) code: A10B)

- combination therapy of OAD and insulin (ATC-codes: A10B and A10A)

- insulin therapy only and ≥50 years

- dietetically treated patients: if 4 or more blood glucose level measurements or two or more HbA1c measurements are documented in the current year.

Pseudonymized data concerning patient's characteristics, prescriptions and main diabetes-relevant admission and discharge diagnoses (ATC-codes and International Classification of Diseases (ICD)-10 codes are presented in S1 Table), number of hospital admissions and days, and costs for in- and outpatient care per calendar year were provided.

## Study cohorts

**Early program cohort.** The evaluation of the long-term follow up of the early program phase was based on the existing propensity score matched cohort described in [15]. The matched cohort includes N = 7 181 DMP participants (newly enrolled in the DMP in 2008/2009) and N = 21 543 controls (patients treated according to usual care). This cohort was evaluated concerning patient-relevant outcomes and economic impact for the observational period of 8 years starting from January 1, 2009 to December 31, 2016 and January 1, 2010 to December 31, 2017 for the two baseline years 2007 and 2008, respectively.

**Established program cohort.** For the evaluation of the established program phase, the same inclusion/exclusion criteria as for the early program phase cohort were applied [15]. In detail: all patients had to be registered in LEICON in the baseline year 2012 (throughout 2016 or deceased). The DMP-group consists of patients newly enrolled in the program between January 1, 2013 and December 31, 2013 and with at least one of the annually planned DMP-documentations by their physicians after enrollment to ensure active participation. The control-group includes patients not enrolled in the DMP before December 31, 2016 and predominantly (more than 80% visits) under treatment of non DMP-physicians. In both groups, patients who died in the following year after baseline were excluded. Patient-relevant outcomes, economic impact costs and quality of care parameters were evaluated for the observational period of 4 years starting from January 1, 2014 to December 31, 2017.

During 2013 N = 5982 patients were enrolled in the DMP. From those, N = 2682 (44.8%) participants were not identified as patients with type 2 diabetes mellitus in the year before in the LEICON database. We refer to this group as "newly registered DMP-group". Characteristics and patient-relevant outcomes, economic impact costs and quality of care parameters were compared descriptively with DMP participants enrolled in 2013 and identified as patients with type 2 diabetes mellitus in 2012 or earlier.

## Endpoints

Patient-relevant outcomes were mortality and major macrovascular diabetes specific complications such as myocardial infarction and stroke/non-traumatic intracranial bleedings. International Classification of Diseases (ICD-10) codes were used to identify myocardial infarction (I21-I22), stroke (I63) and stroke/non-traumatic intracranial bleedings (I60-I64). In the

established program phase only, also amputations were identified by the following individual medical procedure groups (in German: *Medizinische Einzelleistungs-Gruppen*, MEL-groups) [18]: NA070, NZ080, NZ090, NZ100, NZ110, NZ120 and NZ130. For evaluation of the economic impact, total costs (including outpatient physician services costs, hospital costs, prescription costs, transportation costs) and number of hospital admissions and days were investigated.

In the established program cohort only, process quality of care parameters including the number of patients with physician (general practitioner) contacts, eye exams, electrocardiogram performed, and HbA1c testing and other laboratory parameter testing were investigated.

## Statistical analysis

**Propensity score calculation and matching.** In the analysis of the established program phase, DMP participants and controls were matched on their propensity score, defined as probability of DMP participation conditional on baseline covariates [19]. Multivariable logistic regression stratified by the participating regions of Austria (Burgenland, Lower Austria, Upper Austria, Salzburg, Styria, Vorarlberg and Vienna) were used to calculate the PS. The same baseline covariates as in the matching of the early program cohort, (described in detail in [15]) were included in the model: patient's characteristics, form of antidiabetic therapy, the number of hospital admissions and days, costs, several prescriptions and main discharge diagnoses (**S1 Table**). Three controls were matched to one DMP participant based on a nearest-neighbour-matching algorithm without replacement [20]. Absolute standardized differences between the groups were calculated before and after matching to assess the quality of the matching [21].

**Analysis.** In both matched cohorts, Cox-Proportional Hazard Models using a robust sandwich estimator [22] to account for the matched data were used to analyze group differences in all-cause mortality. An observation was censored if the patient was still alive after the follow-up period (8 years and 4 years for the early and established program phase cohort, respectively). The results are presented as hazard ratio (HR) with a 95% confidence interval (CI).

Mean annual total costs per person over 8 years and 4 years were calculated and analyzed via general estimating equations (GEE) models with gamma-distribution and log-link accounting for the matching [23]. Bootstrap-methods were used to calculate 95% CI for the mean annual total cost differences between the groups [24, 25].

Major macrovascular complications, single cost components, the number of hospital admissions and days, quality of care parameters and the newly registered DMP-group were analyzed descriptively. Statistical analysis were performed using SAS Version 9.4.

## Results

### Long-term results of early program cohort

Detailed characteristics of the DMP and control-group before and after matching are published in Riedl [15]. Before matching, patients in the control group tended to be older with a higher number of hospital days and total costs compared to DMP participants. In **Table 1** the characteristics for the DMP-group and the control-group in the early program phase cohort after matching are summarized. Both groups are comparable concerning the matching parameters. In the control-group only 992 (4.6%) of the patients switched to the DMP in the extended follow up period 2014–2017.

**Patient-relevant outcomes.** In the DMP-group 22.1% (1584/7181) of the patients died within the follow-up period of 8 years, whereas in the control-group 29.7% (6387/21543) of the patients died (HR = 0.70; 95% CI: 0.66–0.73). For the major macrovascular diabetic-

**Table 1. Baseline characteristics for the DMP-group and control-group after matching in the early program cohort.**

| | Matched controls N = 21543 | | Matched DMP-group N = 7181 | |
|---|---|---|---|---|
| | N | % | N | % |
| | mean (SD) | median (min-max) | mean (SD) | median (min-max) |
| sex | | | | |
| female | 10953 | 50.8 | 3672 | 51.1 |
| male | 10590 | 49.2 | 3509 | 48.9 |
| age | 64.2 (11.6) | 65 (18–99) | 64 (11) | 65 (18–95) |
| prescription fee | 6711 | 31.2 | 2233 | 31.1 |
| hospital days >0 | 12.9 (14.7) | 8 (1–198) | 13.5 (16.2) | 8 (1–154) |
| hospital admissions >0 | 1.9 (1.9) | 1 (1–41) | 1.8 (1.5) | 1 (1–16) |
| total costs, € | 2746 (3517) | 1598 (7–55420) | 2744 (3654) | 1603 (8–63888) |
| therapy form | | | | |
| none | 3462 | 16.1 | 1112 | 15.5 |
| OAD only | 14302 | 66.4 | 4810 | 67.0 |
| Insulin only | 1508 | 7.0 | 484 | 6.7 |
| combination | 2271 | 10.5 | 775 | 10.8 |

specific complications, slightly lower percentages were observed for the DMP-group with any complication of 10% (719/7161) compared to controls (11.7%, 2461/21044) (**Table 2**).

**Economic impact.** The mean total costs per year amounted € 9859 in the DMP-group and € 10899 in the control-group (p<0.001) with a mean difference of € 1069.90 (95%

**Table 2. Early program phase: Patient-relevant outcomes and economic impact after 8 years follow-up.**

| | DMP-group N = 7181 | | Control-group N = 21543 | |
|---|---|---|---|---|
| **Patient-relevant outcomes** | N | % | N | % |
| Mortality | 1584 | 22.06 | 6387 | 29.65 |
| Hazard Ratio (95% Confidence Interval) | 0.70 (0.66–0.73) | | | |
| Major macrovascular complications[a] | | | | |
| Myocardial infarction (ICD: I21, I22) | 279 | 3.90 | 956 | 4.54 |
| Stroke/non-traumatic intracranial bleedings (ICD: I60-I64) | 464 | 6.48 | 1622 | 7.71 |
| Stroke (ICD: I63) | 312 | 4.36 | 1070 | 5.08 |
| Any complication [b] | 719 | 10.04 | 2461 | 11.69 |
| **Economic impact parameter[a]** | | | | |
| Mean total costs per year | 9 859.70 € | | 10 898.90 € | |
| 95% bootstrap-T interval | 1069.90 € (822.70 € - 1412.40 €) | | | |
| Outpatient physician services costs | 763.50 € | | 702.20 € | |
| Hospital costs | 7 688.60 € | | 8 653.30 € | |
| Prescription costs | 1 333.60 € | | 1 398.70 € | |
| Transportation costs | 93.00 € | | 144.80 € | |
| Hospital admissions and days | | | | |
| Hospital admissions and days >0, N (%) | 6301 (88.0) | | 18218 (86.6) | |
| Cumulative number of hospital days >0 (mean/median) | 48.7/29 | | 51.6/31 | |
| Cumulative number of hospital admissions >0 (mean/median) | 6.9/5 | | 6.8/5 | |

[a] N = 7161 in the DMP-group and N = 21044 in the control-group due to missing values

[b] Included ICD: I21-I22 and/or I60-I64

bootstrap-T interval: € 722.70 - € 1412.40). Slightly higher outpatient physician services costs, lower hospital costs and a lower cumulative number of hospital days over 8 years (median 29 days vs. 31 days) were observed in the DMP-group compared to controls (**Table 2**).

## Results of the established program cohort

In the baseline year 2012, N = 502913 patients with type 2 diabetes mellitus were identified in the database. From those, N = 117062 patients in the control-group and N = 3087 DMP participants fulfilled our inclusion criteria. The 1:3 PS matching yielded N = 3087 DMP participants and N = 9261 controls (**Fig 1**). Before matching, group differences in age (controls were older), number of hospital days and total costs (higher in controls) were observed. The matching resulted in good balance (standardized difference <10%) in all our considered baseline parameters (**Fig 2**). Descriptive statistics for the DMP-group and the control-group after matching in the established program phase cohort are summarized in **Table 3** and **S2 Table**.

**Patient-relevant outcomes.**   Within 4 years after DMP enrollment, 10.6% (327/3087) of the patients died in the DMP-group, whereas in the control-group 11.9% (1099/9261) of the patients died (HR = 0.88, 95% CI: 0.78–0.99, p = 0.038). The major macrovascular diabetes-specific complications are similar in both groups (**Table 4**). Any diabetes-specific included complication was observed in 6.8% (209/3087) DMP-participants and in 7.1% (653/9261) patients in the control-group.

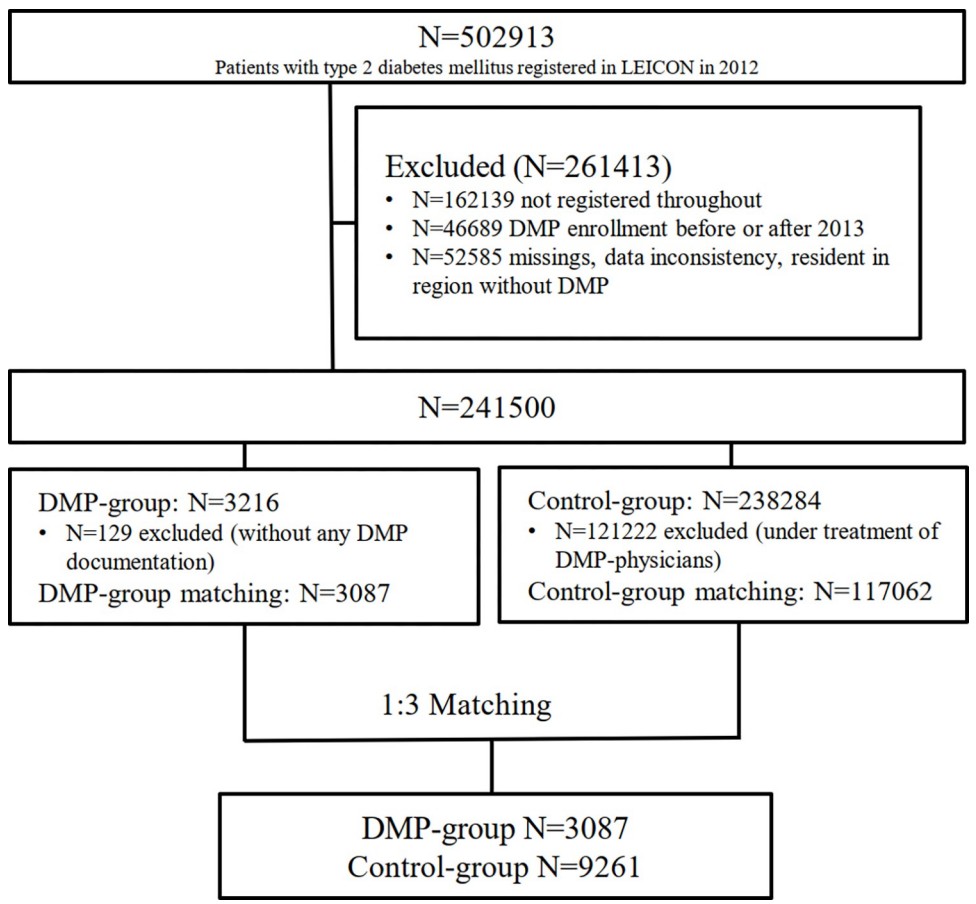

**Fig 1. Flowchart of type 2 diabetic patients considered for matching.**

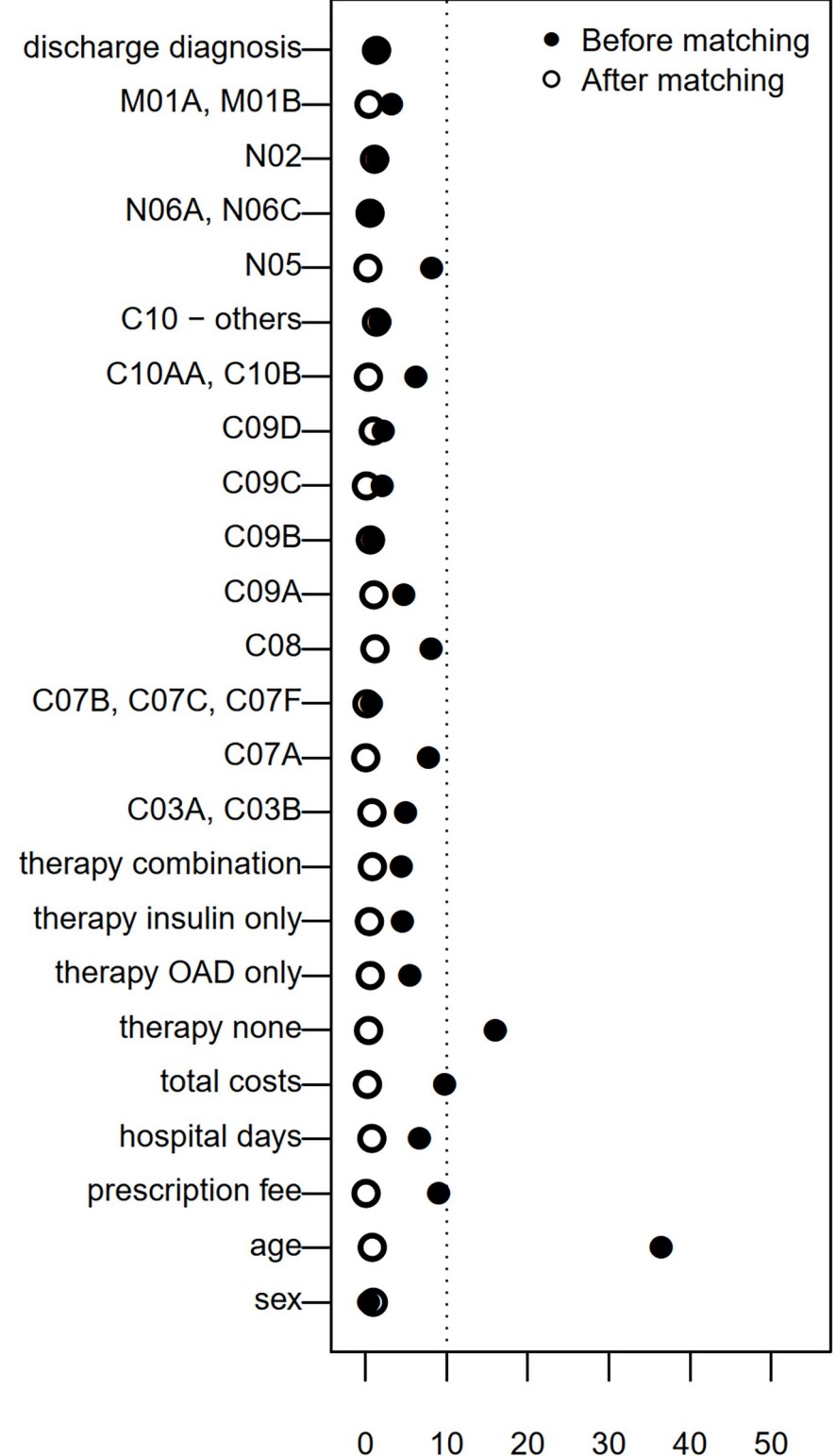

**Fig 2. Absolute standardized differences between DMP-group and control-group before and after matching.**

**Table 3.** Baseline characteristics for the DMP-group and control-group after matching in the established program cohort.

| | Matched controls N = 9261 | | Matched DMP-group N = 3087 | | newly registered DMP-group N = 2225 | |
|---|---|---|---|---|---|---|
| | N mean (SD) | % median (min-max) | N mean (SD) | % median (min-max) | N mean (SD) | % median (min-max) |
| sex | | | | | | |
| female | 4402 | 47.5 | 1482 | 48.0 | 1095 | 49.2 |
| male | 4859 | 52.5 | 1605 | 52.0 | 1130 | 50.8 |
| age | 64 (12) | 64 (18–97) | 64 (12) | 64 (18–95) | 59 (13) | 60 (18–95) |
| prescription fee | 4188 | 45.2 | 1397 | 45.3 | 524 | 23.6 |
| hospital days >0 | 12.4 (15.6) | 7 (1–235) | 12.7 (16.1) | 7 (1–171) | 10.9 (15,9) | 6 (1–158) |
| hospital admissions >0 | 2 (2.5) | 1 (1–83) | 1.9 (1.5) | 1 (1–16) | 1.7 (1.4) | 1 (1–14) |
| total costs, € | 3111 (4427) | 1709 (11–106900) | 3122 (4712) | 1636 (15–90894) | 1621 (3309) | 674 (0–52296) |
| therapy form | | | | | | |
| none | 1343 | 14.5 | 444 | 14.4 | 2225 | 100.0 |
| OAD only | 6304 | 68.1 | 2093 | 67.8 | 0 | 0.0 |
| Insulin only | 687 | 7.4 | 233 | 7.5 | 0 | 0.0 |
| combination | 927 | 10.0 | 317 | 10.3 | 0 | 0.0 |

**Economic impact.** The mean total costs (DMP-group: € 9779, control-group: € 9761, p = 951) and costs components per year (**Table 4**) were similar between the groups. No differences were observed for the cumulative number of hospital days over 4 years, with a median of 16 days in both groups.

**Quality of care parameters.** In the baseline year 2012, quality of care parameters were similar in both groups (physician contacts: about 90%, eye exams and electrocardiogram performed: about 40%, HbA1c and other laboratory parameter testing: about 70%). Over the observational period 2013–2017, the percentage of patients with specific examinations, such as eye exams and HbA1c testing, increased in the DMP-group in the enrollment year 2013 and slightly decreased over time thereafter. For the controls, the quality of care parameters slightly increased over time but remained lower compared with the DMP-group (**Fig 3**). Over the follow up years, 2014–2017, the proportion of patients in the DMP-group is higher in all specified process parameters (75%-96%) compared to the control-group (63%-90%) (**Table 4**).

**DMP participants newly registered in database.** From the N = 2682 newly enrolled and newly registered patients, N = 457 were excluded due to missings, data inconsistency, resident in region without DMP or no DMP-documentation. The descriptive statistics for the remaining N = 2225 DMP participants are summarized in **Table 3**.

Participants newly registered were younger (59±13 years), showed lower total costs (median € 674) and lower hospital days (median 6) compared to the already registered DMP participants. Results for the patient-relevant outcomes, economic impact and quality of care parameter are summarized in **Table 5**. Compared to registered DMP participants, a lower mortality (6.0%, 134/2225), lesser major macrovascular complications and amputations (any complication 3.3%, 73/2212), lower costs (mean total costs per year: € 7048.4) and fewer hospital admissions (median: 2) and days (median: 12) were observed. For quality of care parameters, the proportion of patients varied between 68%-93%.

## Discussion

In this observational study, we updated the first evaluation results for the early program phase for an observational period of 2009/2010–2016/2017 and newly evaluated a cohort in the established program phase with observational period 2014–2017, concerning patient-relevant outcomes, economic impact and quality of care parameters. The long-term results—8 years

**Table 4. Established program cohort: Patient-relevant outcomes, economic impact and quality of care after 4 years follow-up.**

| | DMP-group N = 3087 | | Control-group N = 9261 | |
|---|---|---|---|---|
| **Patient-relevant outcomes** | **N** | **%** | **N** | **%** |
| **Mortality** | **327** | **10.59** | **1099** | **11.87** |
| Hazard Ratio (95% Confidence Interval) | 0.88 (0.78–0.99) | | | |
| Major macrovascular complications and amputations[a] | | | | |
| Myocardial infarction (ICD-10: I21, I22) | 76 | 2.47 | 267 | 2.91 |
| Stroke/non-traumatic intracranial bleedings (ICD-10: I60-I64) | 115 | 3.74 | 331 | 3.61 |
| Stroke (ICD-10: I63) | 78 | 2.53 | 214 | 2.33 |
| Amputations (MEL: NA070, NZ080, NZ090, NZ100, NZ110, NZ120 und NZ130) | 29 | 0.94 | 91 | 0.99 |
| Any complication [b] | 182 | 5.91 | 580 | 6.31 |
| Any complication including amputations [c] | 209 | 6.79 | 653 | 7.11 |
| **Economic impact parameter[a]** | | | | |
| Mean total costs per year | 9 779.30 € | | 9 761.00 € | |
| 95% bootstrap-T interval | -21.20 € (-607.20 € - 525.60 €) | | | |
| Outpatient physician services costs | 811.40 € | | 714.50 € | |
| Hospital costs | 7 538.30 € | | 7 520.50 € | |
| Prescription costs | 1 328.20 € | | 1 414.50 € | |
| Transportation costs | 101.40 € | | 111.50 € | |
| Hospital admissions and days | | | | |
| Hospital admissions and days >0, N (%) | 2210 (71.8) | | 6425 (69.9) | |
| Cumulative number of hospital days >0 (mean/median) | 30.3/16 | | 29.5/16 | |
| Cumulative number of hospital admissions >0 (mean/median) | 4.5/3 | | 4.3/3 | |
| **Quality of care parameter[a]** | **N** | **%** | **N** | **%** |
| Physician contacts (general practitioner) | 2959 | 96.2 | 8656 | 94.2 |
| Eye exams | 2321 | 75.4 | 5977 | 65.0 |
| Electrocardiogram performed | 2393 | 77.8 | 5792 | 63.0 |
| HbA1c testing | 2952 | 95.9 | 8136 | 88.5 |
| Other laboratory parameter testing | 2938 | 95.5 | 8327 | 90.6 |

[a] N = 3077 in the DMP-group and N = 9190 in the control-group due to missing values

[b] Included ICD-10: I21-I22 and/or I60-I64

[c] Included ICD-10: I21-I22 and/or I60-I64 and/or amputations

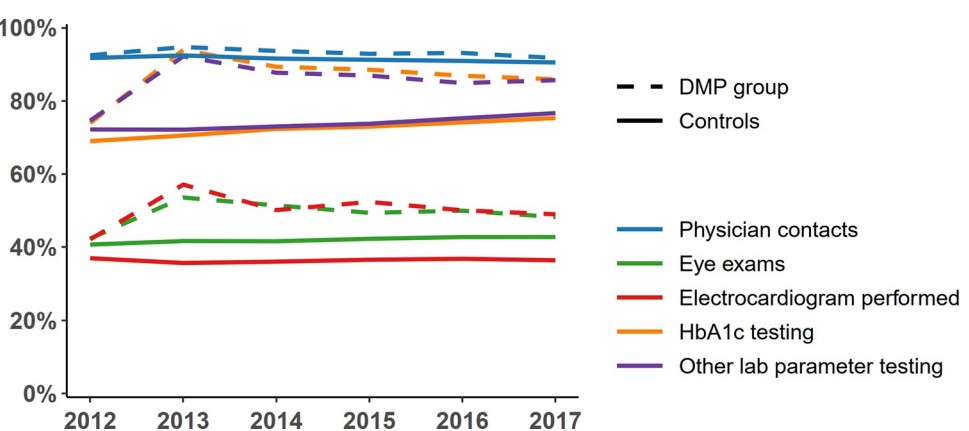

**Fig 3. Quality of care parameters for DMP group and controls from 2012 to 2017.**

**Table 5. Established program phase: Results for the newly registered DMP-group after 4 years follow-up.**

|  | newly registered DMP-group N = 2225 | |
|---|---|---|
| **Patient-relevant outcomes** | N | % |
| Mortality | 134 | 6.02 |
| Major macrovascular complications and amputations[a] | | |
| Myocardial infarction (ICD-10: I21, I22) | 19 | 0.86 |
| Stroke/non-traumatic intracranial bleedings (ICD-10: I60-I64) | 50 | 2.26 |
| Stroke (ICD-10: I63) | 28 | 1.27 |
| Amputations (MEL: NA070, NZ080, NZ090, NZ100, NZ110, NZ120 und NZ130) | 5 | 0.23 |
| Any complication [b] | 69 | 3.12 |
| Any complication including amputations [c] | 73 | 3.30 |
| **Economic impact parameter[a]** | | |
| Mean total costs per year | 7 048.40 € | |
| Outpatient physician services costs | 655.00 € | |
| Hospital costs | 5 605.90 € | |
| Prescription costs | 738.60 € | |
| Transportation costs | 48.90 € | |
| Hospital admissions and days | | |
| Hospital admissions and days >0, N (%) | 1346 (60.8) | |
| Cumulative number of hospital days >0 (mean/median) | 25.1/12 | |
| Cumulative number of hospital admissions >0 (mean/median) | 3.6/2 | |
| **Quality of care parameter[a]** | N | % |
| Physician contacts (general practitioner) | 2059 | 93.1 |
| Eye exams | 1512 | 68.4 |
| Electrocardiogram performed | 1587 | 71.7 |
| HbA1c testing | 2016 | 91.1 |
| Other laboratory parameter testing | 2046 | 92.5 |

[a] N = 2212 in the DMP-group due to missing values

[b] Included ICD-10: I21-I22 and/or I60-I64

[c] Included ICD-10: I21-I22 and/or I60-I64 and/or amputations

follow-up—of the early program cohort showed that the association between DMP participation and the two endpoints remains i.e. lower mortality and lower total costs, primarily due to the higher hospital costs in the control-group. In our established program cohort, a lower mortality rate, similar costs, and a better quality of care for the DMP participants in comparison to the control-group were observed.

Comparing the results of the established cohort with our first evaluation with an observational period of 2009/2010–2012/2013 [15], we observed similar baseline characteristics (including sex, age, hospital days, costs and form of antidiabetic drug therapy) in both matched cohorts (early and established, **Tables 1** and **3**). However, in the established program cohort, we found a lower effect for mortality and major macrovascular diabetes complications. After four years follow up, the HR in the established program cohort was HR = 0.88 (95% CI: 0.78–0.99) and HR = 0.57 (95% CI: 0.52–0.61) in the early program cohort. The mortality for DMP participants remained similar (early 9.4% vs established 10.6%) across cohorts, whereas the mortality in the control-group changed from 15.9% to 11.9%. This reduction may be explained by the following aspects: First, medical care improved over time [26–28]. The development of new classes of diabetes drugs in the recent years reduced the risk for diabetes-specific

complications [26]. Moreover, adequate control of risk factors can reduce the diabetes-associated risk for death, stroke, and myocardial infarction to only a marginally higher risk compared to the general population [28]. These improvements in medical care likely reduce the net effect of the training (for physicians and patients) and the treatment targets as key components in the DMP. However, mortality in the DMP group remained similar across the cohorts included in the matchings. Similar baseline characteristics were observed as well. Due to the lack of data on disease duration and HbA1c measurements, it cannot be ruled out that the cohorts might differ in disease severity, which could explain the lack of improvement across the DMP cohorts. Second, interest in diabetes and support for patients notably improved in Austria in the last decade [3, 29]. This might have increased the knowledge and health awareness of diabetes in general. Third, positive spillover effects [30] might be present: non-participants, including patients and physicians, can indirectly profit from the program. For example, the diabetes training is also offered to patients who do not want to enroll in the DMP [31]. We may speculate, that spillover effects are more present in the established phase of the DMP due to the growing awareness of the program in Austria. While we tried to reduce the influence of spillover effects by excluding patients in the control-group predominantly under treatment of DMP physicians, we cannot rule out the influence of such effects on non-DMP physicians.

Despite the mentioned improvements in diabetes care, the observed positive effects of the DMP in our early program cohort persisted over 8 years follow up. This may be explained by the importance of early optimal diabetes control and risk factor management to delay progression and prevent complications [12, 32, 33].

In our established program cohort, we observed better process quality of care for DMP participants over 2013–2017. Better quality of care for Austrian DMP participants compared to non-participants was also observed by Sönnichsen et al [17] and Ostermann et al [11]. In accordance with Osterman et al in 2009, the largest differences (about 20% difference) between the two groups were seen in percentage of patients with HbA1c measurements and other laboratory parameter testing in the enrollment year 2013 of the DMP. Although higher than in non-participants, the yearly-recommended eye and electrocardiogram examinations only were performed in about 50% of DMP participants in the single years.

The DMP has been adapted over time with regard to the documentation sheet and treatment paths based on current guidelines. A comprehensive update (including the structured training of patients, diagnosis of diabetes, adoption of HbA1c and LDL target values) was performed in 2015 [34]. More importantly, the enrollment behavior changed over time. There is a strong tendency to include patients in the program as early as possible. From 2007 to 2013, the median age and diabetes duration at enrollment decreased from 66 to 63 years and 6 to 3 years, respectively. This may explain why about 45% of DMP participants were not identified as type 2 diabetic patients in LEICON in 2012.

The main goal of implemented DMPs in Europe is to improve the quality of chronic healthcare and thereby improve patient outcomes including quality of life and lower diabetes-related complications and extend their lifetime in good health. A comprehensive overview of the key features of DMPs for diabetes and their evaluations in Europe is given in Kostial et al. [6]. Due to large heterogeneity in evaluation methodology, direct comparisons of the results are difficult. However, similar to the DMP in Austria is the DMP in Germany [15]. Beneficial impact on mortality but inconsistent effects on morbidity, quality of life and economic parameters were observed by Fuchs et al. [35] in earlier program evaluations. More recently, supporting our results for process quality of care, Mehring et al. [10] observed an improvement in quality of care, patient education and therefore improved adherence to guidelines over time in diabetic DMP participants in Germany. A better quality of care was also observed by Höglinger et al. [36] who investigated the impact of DMPs in Switzerland on similar quality of care

parameters as in our study. In their evaluation, patients with diabetes mellitus type 1 and type 2 and treated with antidiabetic medication were included and better guideline-adherent care (including measurement of HbA1c, lipid profile, nephropathy status and examinations by the ophthalmologist) was observed in a follow-up period of two years.

Our study has several limitations: about 45% of newly enrolled DMP participants in 2013 were not included in the matched analysis. Hence, our matched cohort is not representative for all DMP participants and the generalizability of our results may be limited especially for prediabetic patients and patients with short disease duration. Including all DMP participants (i.e. ignoring the inclusion criteria that a patient has to be registered in LEICON in the baseline year 2012) in a sensitivity analysis, resulted in bad balance after 1:1 matching due to small overlap between the groups. A further limitation represents the fact that important clinical parameters like HbA1c measurements and duration of diabetes were not available. This might affect the comparability of our groups due to different baseline clinical conditions (unmeasured confounding), and allows only general statements about effectiveness of the program. It might be possible that patients with more advanced disease stage and/or having more difficulties in controlling diabetes relevant clinical parameters are included in the DMP group of the established program cohort. However, due to the lack of clinical data, this cannot be verified. As discussed in Riedl et al [15], misclassification of diabetic patients might be possible. We think that the inclusion of patients who fulfilled the algorithm for diabetes identification from baseline throughout 2016 (if not deceased), reduces the risk of this misclassification. Although, in our matching approach resulted in good balance, we cannot rule out the influence of residual confounding and unmeasured confounding including higher motivation of patients and physicians or other lifestyle related factors.

Strengths of the study are the large population based cohort design with a broad consideration of matching variables and the evaluation of long-term results of patients included in the early phase of the DMP as well as results in an established phase of the program. Especially for the latter not much information is provided.

## Conclusions

In conclusion, we observed lower mortality rates and better process quality of care for DMP participants compared to non-participants. The fact that mortality rate decreased for non-participants as well, is an indirect indication that "usual care" for type 2 diabetic patients improved to "better care" through time.

## Supporting information

**S1 Table. List of included prescriptions based on Anatomical Therapeutic Chemical (ATC) classification system and discharge diagnoses based on International Classification of Diseases (ICD10) codes.**
(DOCX)

**S2 Table. Established program cohort: Descriptive statistics for the DMP-group and the control-group before and after matching.**
(DOCX)

## Acknowledgments

We thank Dr. Nia Owen (Institute for Medical Informatics, Statistics and Documentation) for supporting the literature update.

## Author Contributions

**Conceptualization:** Regina Riedl, Andrea Berghold.

**Data curation:** Martin Robausch.

**Formal analysis:** Regina Riedl.

**Supervision:** Andrea Berghold.

**Writing – original draft:** Regina Riedl.

**Writing – review & editing:** Regina Riedl, Martin Robausch, Andrea Berghold.

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
