## [Decision Letter · Decision Letter 0]

9 Sep 2022

PONE-D-22-21147Disease Management Program in patients with type 2 diabetes mellitus, long-term results of the early and established program cohort: a population-based retrospective cohort studyPLOS ONE

Dear Dr. Berghold,

Thank you for submitting your manuscript to PLOS ONE. After careful consideration, we feel that it has merit but does not fully meet PLOS ONE’s publication criteria as it currently stands. Therefore, we invite you to submit a revised version of the manuscript that addresses the points raised during the review process.

We look forward to receiving your revised manuscript.

Kind regards,

Gianluigi Savarese

Academic Editor

PLOS ONE

Journal Requirements:

 "This study was funded by the Österreichische Gesundheitskasse/Versorgungsmanagement 3 in charge of the DMP “Therapie aktiv –   Diabetes im Griff”. There was no involvement in study design; in the collection,         analysis and interpretation of data; in the   writing of the report; and in the decision to submit the article for publication."

Please state what role the funders took in the study.  If the funders had no role, please state: ""The funders had no role in study design, data collection and analysis, decision to publish, or preparation of the manuscript."" If this statement is not correct you must amend it as needed. 

"We thank Mag. Heinrich Koch and Mag. Helmut Nagy from the Österreichische Gesundheitskasse/Versorgungsmanagement 3 for   providing detailed information on the DMP “Therapie Aktiv” and Dr. Nia Owen (Institute for Medical Informatics, Statistics and   Documentation) for supporting the literature update."

"This study was funded by the Österreichische Gesundheitskasse/Versorgungsmanagement 3 in charge of the DMP “Therapie aktiv – Diabetes im Griff”. There was no involvement in study design; in the collection, analysis and interpretation of data; in the writing of the report; and in the decision to submit the article for publication."

"Martin Robausch is project leader of the LEICON database, working at the Lower Austria Health Insurance Fund, who is   responsible for the implementation of the DMP in Lower Austria. All other authors declare that they have no conflict of  interest."

Reviewers' comments:

Reviewer's Responses to Questions

**Comments to the Author**

1. Is the manuscript technically sound, and do the data support the conclusions?

Reviewer #1: Yes

Reviewer #2: Partly

2. Has the statistical analysis been performed appropriately and rigorously? 

Reviewer #1: N/A

Reviewer #2: Yes

3. Have the authors made all data underlying the findings in their manuscript fully available?

Reviewer #1: Yes

Reviewer #2: No

4. Is the manuscript presented in an intelligible fashion and written in standard English?

Reviewer #1: Yes

Reviewer #2: Yes

5. Review Comments to the Author

Reviewer #1: Thank you submitting this interesting paper exploring the effect of the DMP program on mortality, diabetes-related complications, costs and process quality of care parameters in the Austrian T2DM population.

The paper is overall well written and worth of attention, as highlighting the importance of dedicated care and dedicated programs to tackle diseases (T2DM in this specific case) that nowadays still present with incredibly high morbidly and mortality rates notwithstanding the extensive research in both terms of prevention and treatment.

In my opinion, however, the lack of data concerning disease duration and values of HbA1 values represents a strong limitation to the interpretation of results. In fact, even if the improvement in general care may explain the reduction in mortality and diabetes-specific complications HR in the control group, this does not fully explain why figures do not show the same trend in the DMP group as well. Could this be explained by patients enrolled in the DMP program being in more advanced stages of disease or having more difficulties in controlling DM parameters, as a consequence being more motivated in taking part to the program?

In this scenario, the beneficial effect of the program may be not extended to the whole population but, on the contrary, to a specific part of it with specific inclusion criteria and the adherence to program itself be biased by baseline clinical conditions.

I would maybe consider and expand this point more.

Reviewer #2: In this observational case-control study, the authors investigate the effectiveness of the Disease Management Program “Therapie aktiv – Diabetes im Griff”, a systematic treatment program for patients with diabetes mellitus type 2. They investigate the long-term effects of the program for participants enrolled in the early phase (2008-2009) and evaluated a cohort enrolled later, in the established phase of the program, adding patient-relevant outcomes (overall mortality, cardiovascular disease) and quality of care.

The main findings are that patients included in the early phase of the program have lower mortality rates and lower health-related costs, mainly due to the lower rate of hospitalizations. Patients newly included had lower mortality and a better quality of care compared with controls, but the costs were similar. In particular, the authors observe that mortality remained similar in patients enrolled in the program (both early phase and established phase), but in the control groups, which are different for the two cohorts, mortality decreased significantly.

The manuscript reports interesting results, however some concerns should be addressed:

- It would be useful for the reader to have a more detailed description of the program and what it encompasses. How are patients enrolled and how do physicians choose to participate?

- The results do not seem to be thoroughly discussed. There is no reference to similar programs in other countries. While they might be quite different because of diverse healthcare systems, it might be meaningful to discuss the goal of such programs.

- How were diabetes complications chosen? Macrovascular disease seems to be quite well represented but kidney disease and retinopathy are not included. Moreover, one could evaluate including heart failure hospitalizations, as they presumably have a significant impact on health costs and might be related to poor diabetes control.

- While it is understandable that one uses PS matching to investigate outcomes, in a study having a descriptive part one might be interested in the comparison of patients included in the program with those not included in general. Which are the factors that make it more likely to be included vs not included?

Minor issues:

- Mismatch in references: for example, it is stated that Riedl et al is ref 15 whereas it is ref 16

- References 1 and 2 should be updated with new data form more recent reports

6. PLOS authors have the option to publish the peer review history of their article (what does this mean?). If published, this will include your full peer review and any attached files.

Reviewer #1: No

Reviewer #2: No

---

## [Author Response · Author response to Decision Letter 0]

19 Oct 2022

We answered all comments in the cover letter and response to reviewers file.

---

## [Decision Letter · Decision Letter 1]

1 Dec 2022

Disease Management Program in patients with type 2 diabetes mellitus, long-term results of the early and established program cohort: a population-based retrospective cohort study

PONE-D-22-21147R1

Dear Dr. Berghold,

We’re pleased to inform you that your manuscript has been judged scientifically suitable for publication and will be formally accepted for publication once it meets all outstanding technical requirements.

Kind regards,

Gianluigi Savarese

Academic Editor

PLOS ONE

Additional Editor Comments (optional):

Reviewers' comments:

Reviewer's Responses to Questions

**Comments to the Author**

1. If the authors have adequately addressed your comments raised in a previous round of review and you feel that this manuscript is now acceptable for publication, you may indicate that here to bypass the “Comments to the Author” section, enter your conflict of interest statement in the “Confidential to Editor” section, and submit your "Accept" recommendation.

Reviewer #1: All comments have been addressed

Reviewer #2: All comments have been addressed

2. Is the manuscript technically sound, and do the data support the conclusions?

Reviewer #1: Yes

Reviewer #2: Yes

3. Has the statistical analysis been performed appropriately and rigorously? 

Reviewer #1: N/A

Reviewer #2: Yes

4. Have the authors made all data underlying the findings in their manuscript fully available?

Reviewer #1: Yes

Reviewer #2: No

5. Is the manuscript presented in an intelligible fashion and written in standard English?

Reviewer #1: Yes

Reviewer #2: Yes

6. Review Comments to the Author

Reviewer #1: Thank you for amending accordingly. Hopefully, the possibility in the future to add the missing data may improve our understanding in the field further.

Reviewer #2: The authors have addressed the comments at their best.

I suggest to add in the discussion references to the EUROASPIRE program for the years you considered, specifically on diabetes screening and management. I suggest references PMID: 29368616 and PMID: 32079627.

7. PLOS authors have the option to publish the peer review history of their article (what does this mean?). If published, this will include your full peer review and any attached files.

Reviewer #1: No

Reviewer #2: No

---

## [Editor Report · Acceptance letter]

5 Dec 2022

PONE-D-22-21147R1 

Disease Management Program in patients with type 2 diabetes mellitus, long-term results of the early and established program cohort: a population-based retrospective cohort study 

Dear Dr. Berghold:

I'm pleased to inform you that your manuscript has been deemed suitable for publication in PLOS ONE. Congratulations! Your manuscript is now with our production department. 

Kind regards, 

on behalf of

Dr. Gianluigi Savarese 

Academic Editor

PLOS ONE